# Rhamnan Sulfate from the Seaweed *Monostroma nitidum* May Improve Cognitive Impairment Caused by Hyperglycemia

**DOI:** 10.3390/md23060250

**Published:** 2025-06-12

**Authors:** Takaichi Miwa, Masaya Sato, Ning Ma, Keiichi Hiramoto, Masahiro Terasawa, Koji Suzuki

**Affiliations:** 1Faculty of Pharmaceutical Sciences, Suzuka University of Medical Science, Minamitamagakicho 3500-3, Suzuka 513-8670, Mie, Japan; yakuzai@hinaga.or.jp (M.S.); hiramoto@suzuka-u.ac.jp (K.H.); 2Department of Pharmacy, Hinaga General Center for Mental Care, Hinaga 3500, Yokkaichi 510-8575, Mie, Japan; 3Graduate School of Health Science, Suzuka University of Medical Science, Kishioka-cho 1275-3, Suzuka 510-0226, Mie, Japan; maning@suzuka-u.ac.jp; 4Konan Chemical Manufacturing Co., Ltd., Kitagomizuka, Kusu-cho, Yokkaichi 510-0103, Mie, Japan; terasawa@konanchemical.co.jp

**Keywords:** diabetic cognitive impairment, hippocampal atrophy, microglia, rhamnan sulfate, *Monostroma nitidum*

## Abstract

Rhamnan sulfate (RS), extracted from the seaweed *Monostroma nitidum*, suppresses vascular endothelial inflammation and arteriosclerosis, decreases blood glucose levels, and improves blood lipid metabolism and the intestinal environment. We examined whether RS improves hyperglycemia-induced cognitive decline in a hyperglycemic mouse model pretreated with nicotinamide and streptozotocin and then orally administered a high-fat diet and maltodextrin (MD) for 4 months. RS was administered in an MD solution at doses of 75, 225, and 750 mg/kg of mouse body weight. Administration of RS to hyperglycemic mice significantly reduced blood glucose levels and tended to improve memory function in behavioral pharmacological tests using spontaneous locomotor activity, rotarod test, and eight-way-maze test, although the results were not significant. Brain histopathological analysis showed that RS significantly reduced atrophy of neuronal layers in each region of the hippocampus compared with untreated hyperglycemic controls. RS also significantly suppressed TNF-α expression and microglial activation in the brain. These results suggest that RS intake suppresses inflammation in the brain and alleviates the cognitive impairment associated with hyperglycemic diabetes.

## 1. Introduction

In recent years, the incidence of diabetes, which is closely related to lifestyle habits, has been increasing year by year. Diabetes is not limited to middle-aged and elderly people, but is also increasing in younger people. The ACCORD-MIND study has demonstrated that diabetes is an important risk factor for dementia, including Alzheimer’s disease [1]. Meta-analyses [2,3] and cohort studies [4,5], including the Hisayama-cho study [6], have shown that patients with diabetes are at a higher risk of developing Alzheimer’s disease than people without diabetes. Recent studies have revealed that some cases of dementia in hyperglycemia and diabetes are caused by advanced glycation end products (AGEs) and brain inflammation leading to cognitive impairment [7], increased oxidative stress, and insufficient insulin action in the brain due to insulin resistance [8,9].

Previous studies have shown that long-term hyperglycemia causes cognitive impairment, AGE accumulation in the brain, and hippocampal atrophy [10]. This suggests that hyperglycemia affects cognitive impairment and hippocampal atrophy through a specific mechanism. However, studies conducted to date have not yet provided conclusive evidence on the causal relationship between elevated blood glucose levels and cognitive decline.

Therefore, in order to clarify the relationship between diabetes and dementia and its underlying causes, many studies have been conducted to develop preventive measures such as treatments for diabetic dementia using the PPAR-γ agonist pioglitazone [11], dipeptidyl peptidase-4 inhibitors [12], sulfated seaweed polysaccharides [13], and strict control of postprandial blood glucose levels [14]. However, most studies to date have not conclusively proven that these approaches are effective in preventing or treating cognitive decline.

Seaweeds, which are traditionally consumed in East Asia, including Japan, are rich in essential nutrients, including iodine, iron, vitamin C, vitamin K, and vitamin B12, and are a natural source of antioxidants, soluble and insoluble dietary fiber, and polysaccharides [15]. Previous research has revealed that seaweed is a functional food with health-promoting properties, and among the physiologically active components of seaweed, rhamnan sulfate (RS), a polysaccharide extracted from the green algae *Monostroma (M.) nitidum*, has been found to have various health-promoting properties. RS is a soluble fiber found in the intercellular substance of *M. nitidum* and is composed mainly of linear rhamnose chains with side chains and approximately 25% sulfate groups [16,17]. We have previously demonstrated that RS has numerous beneficial effects, including the suppression of blood glucose levels [18], anti-obesity effects [18], improvement in intestinal environment [18], anti-inflammatory effects on vascular endothelial cells [19] and the whole body [20], and anti-arteriosclerosis effects through improvement in lipid metabolism and suppression of vascular inflammation [21], suggesting its potential to prevent various lifestyle-related diseases. Furthermore, we recently demonstrated that RS administration in mice promoted the formation of gut microbiota that is beneficial for health promotion [22].

In this study, we investigated whether RS, which has been shown in previous studies to suppress blood glucose levels and have anti-inflammatory effects on vascular endothelial cells, is effective in improving cognitive impairment and brain tissue damage in a hyperglycemic mouse model.

## 2. Results

### 2.1. Body Weight and Blood Glucose Levels

As shown in Figure 1A, the body weight of the MD group gradually increased during the 4-month study period (1.4 to 1.8 times that of the control), and showed a significant weight gain compared with the control group (*p* < 0.001). No significant changes were observed in the RS-administered groups compared with the MD group.

As for blood glucose levels, as shown in Figure 1B, after 4 months, blood glucose levels in the MD group increased significantly compared with the control group (*p* < 0.0001). Meanwhile, all groups administered RS (MD + RS75, *p* < 0.01; MD + RS225, *p* < 0.001; MD + RS750, *p* < 0.001) showed significantly decreased blood glucose levels compared to the MD group. These results indicate that RS effectively suppresses the rise in blood glucose levels in this model.

### 2.2. Memory Function in Behavioral Pharmacological Tests

#### 2.2.1. Effect of RS on Spontaneous Locomotor Activity

To evaluate the effect of RS on locomotor activity, we measured the number of spontaneous movements. The results are shown in Figure 2. There was no statistically significant difference in locomotor activity between the control and MD-fed groups, and all conditions showed similar activity levels (0.75 to 0.86 times higher than the control group).

#### 2.2.2. Effect of RS on Rotarod Test

The fall latency was measured using the rotarod test to investigate the effects of RS administration on locomotor function. The fall latency values are shown in Figure 3. After 4 months of hyperglycemia, the MD group showed a significantly reduced fall latency compared to the control group (*p* < 0.001). The RS group tended to have a longer fall latency than the MD group, but due to the large variability, this difference was not statistically significant.

#### 2.2.3. Effect of RS on Eight-Way Maze Test

To investigate the effects of RS administration on motor function and memory in cognitive function, task completion time and number of errors were measured using the eight-way maze test. The time required to complete the task is shown in Figure 4A, and the number of errors is shown in Figure 4B. The MD group showed a significant increase in task completion time compared to the control group (*p* < 0.0001), suggesting that glucose loading impairs motor and cognitive functions. No significant differences were observed in task completion times between the RS-administered groups and the MD group. The error rate of the MD group was also significantly higher than that of the control group (*p* < 0.0001). The mean error rate of the MD + RS75 group was reduced by 67% compared to the MD group, but there was a high degree of variability and this reduction was not statistically significant (*p* = 0.061). These results suggest that RS administration did not have a significant effect on improving memory in this experimental model.

### 2.3. Analysis of Hippocampal Pathology Specimens

#### 2.3.1. Effect of RS on Hippocampal Neuronal Layers (CA1, CA2, CA3, and Dentate Gyrus)

To investigate the effect of RS administration on hippocampal structure, the thicknesses of the CA1, CA2, CA3, and dentate gyrus regions of the hippocampal neuronal layer were compared. As shown in Figure 5, the thicknesses of the CA1, CA2, CA3, and dentate gyrus regions were significantly decreased in the MD group compared to those in the control group (CA1, *p* < 0.0001; CA2, *p* < 0.0001; CA3, *p* < 0.0001; dentate gyrus, *p* < 0.0001). In contrast, RS treatment significantly improved the thickness of these regions. The MD + RS75 and MD + RS225 groups showed statistically significant improvements compared to the control group for all thicknesses of the CA1, CA2, CA3, and dentate gyrus regions (MD + RS75/CA1: *p* < 0.0001; CA2, *p* < 0.0001; CA3, *p* < 0.01; dentate gyrus, *p* < 0.0001) (MD + RS225/CA1: *p* < 0.0001, CA2: *p* < 0.0001, CA3: *p* < 0.05, dentate gyrus: *p* < 0.0001). In the MD + RS750 group, the thickness of the CA2 region improved significantly (CA2: *p* < 0.05), and the thicknesses of the CA1, CA3, and dentate gyrus regions showed a trend toward improvement; however, the difference did not reach statistical significance (*p* < 0.1 each). These results indicate that RS treatment suppresses hippocampal atrophy induced by glucose loading and partially restores the thickness of the hippocampal neuronal layer.

#### 2.3.2. Effect of RS on TNF-α Expression in the Brain

Since activated microglial cells in the brain are known to secrete various inflammatory cytokines, we measured the expression levels of TNF-α, a representative cytokine produced by microglial cells, in the brain tissue of mice from the control, MD, and MD + RS groups. TNF-α was evaluated by immunohistochemical staining using an anti-TNF-α antibody. The images of the TNF-α expression levels in each group are shown in Figure 6. In addition, a comparison of TNF-α expression levels between groups based on staining intensity in each group is shown in Figure 7. TNF-α expression was low in the control group, but significantly increased in the MD group compared to the control group (*p* < 0.01). In the RS-administered group, the increase in TNF-α expression observed in the MD group was suppressed. The average staining level of TNF-α was reduced by approximately one-quarter in the MD + RS225 group and approximately one-third in the MD + RS750 group compared with the MD group. Statistical analysis showed that, compared with the MD group, the expression levels of TNF-α were significantly decreased in the MD + RS225 group (*p* < 0.01) and MD + RS750 group (*p* < 0.05), although there was some variability. These results suggest that RS administration suppresses TNF-α expression in the brain under hyperglycemic conditions and reduces inflammation-associated brain damage.

#### 2.3.3. Effect of RS on Iba1 Expression in the Brain

To investigate the effect of RS on microglial cell activation related to brain inflammation, we stained the cells with an antibody against ionized calcium-binding adapter molecule 1 (Iba1), a microglial activation marker [23], and compared its expression levels between groups. The stained images are shown in Figure 8. The expression levels of Iba1 were quantified from the stained images, and the results are shown in Figure 9. In the MD group, Iba1 expression was significantly higher than that in the control group (*p* < 0.0001). In all groups administered RS, the increased expression of Iba1 observed in the MD group was effectively suppressed. Statistical analysis showed that Iba1 expression was significantly suppressed in the MD + RS75, MD + RS225, and MD + RS750 groups compared to that in the MD group (all *p* < 0.0001). These results suggest that RS suppressed cerebrovascular inflammation associated with glucose loading, thereby suppressing microglial cell activation in brain tissue.

## 3. Discussion

In this study, we investigated the effects of RS administration for 4 months on blood glucose levels, cognitive function, and brain tissue lesions in an MD-induced hyperglycemic diabetic mouse model.

As previously reported [18], we confirmed that the administration of RS significantly suppressed the increase in blood glucose levels in MD-treated hyperglycemic mice.

Cognitive function was comprehensively evaluated from three behavioral pharmacology tests: a spontaneous locomotor test to measure the activity level, a rotor rod test to evaluate motor dysfunction in cognitive function, and an eight-way maze test to evaluate memory in cognitive function. Behavioral pharmacology tests showed that no significant differences in locomotor activity were observed between the MD and control groups, with or without RS administration, indicating that neither MD nor RS had a direct effect on locomotor function. In the rotor rod test, the MD group showed a significant decrease in fall latency compared with the control group. Although not statistically significant, the RS group tended to have a longer fall latency in the rotor rod test than the MD group. In the eight-way maze test, the RS group tended to make fewer errors than the MD group. Because RS is a functional food, it is thought that long-term intake is necessary, and in two studies evaluating cognitive function, no statistically significant differences was observed, but a tendency toward improvement in cognitive function was observed. This suggests that RS may contribute to improving cognitive function, but further verification by long-term confirmation is required.

The histopathological studies of the brain suggested that significant atrophy was observed in all regions of the hippocampus (CA1, CA2, CA3, and dentate gyrus) in the MD group, whereas the RS group (75 mg/kg or 225 mg/kg) showed significantly suppressed glucose loading-induced hippocampal atrophy and partially restored the thickness of the hippocampal neuronal layer.

In the hyperglycemic diabetic model mice used in this study, it has been suggested that hyperglycemia induces pancreatic β cell death and that the inflammatory cytokines such as TNF-α secreted from pancreatic β cells migrate to the brain [24]. It is unclear whether the increased TNF-α in the brain of the hyperglycemic mice originates from β cells caused by pancreatitis or from microglial cells due to inflammation in the brain. However, in fact, hyperglycemia in the blood and significant TNF-α production in the brain were observed in the MD group, whereas blood glucose levels decreased and TNF-α levels in the brain were significantly reduced in the RS-administered group, which may be due to the suppression of pancreatitis. We previously reported that RS directly suppresses TNF-α-induced inflammation in cultured vascular endothelial cells [18,19]. Therefore, our findings suggest that RS suppresses inflammation in vascular endothelial cells in organs including the pancreas, suppresses TNF-α-induced pancreatic β cell damage, improves insulin secretion from β cells, and suppresses the increase in blood glucose levels. Additionally, as RS also has an α-amylase activity inhibitory effect [18], RS intake could suppress the absorption of MD–glucose, thereby suppressing the increase in postprandial blood glucose levels.

Iba1 is a cytoskeleton protein specific to microglia and macrophages and is involved in the remodeling of the actin cytoskeleton in the central nerve system [23,25]. It is also known as a microglial activation marker that increases during inflammatory responses and the exacerbation of neurological diseases [23,25]. In this study, its expression was specifically increased in the MD group. This indicates that inflammation occurs in the brain under hyperglycemia and is thought to induce cognitive dysfunction. Furthermore, this study showed that RS administration significantly reduced Iba1 expression in the brain, suggesting that RS suppressed microglial cell activation during glucose loading-induced brain inflammation. As the increased expression of TNF-α in the brain may be derived from activated microglial cells, the decreased expression of TNF-α by RS administration may be due to the inhibition of microglial cell activation in the brain tissue. As shown by the increase in Iba1 in the brains of diabetic mice, the activation of brain microglial cell is closely related to the inflammation of brain endothelial cells [26], and the inhibitory effect of RS administration on the activation of brain microglial cells suggests that RS may suppress inflammation of brain endothelial cells. Anti-inflammatory low-molecular-weight mediators, such as nitric oxide, are presumed to pass through the blood–brain barrier (BBB) and may be involved in the suppression of inflammation in brain endothelial cells.

AGEs are produced in the blood under chronic hyperglycemic conditions and are known to impair the function of endothelial cells in blood vessels throughout the body. It has been suggested that AGEs damage the BBB by injuring cerebrovascular endothelial cells [27]. When the BBB is impaired, high-molecular-weight inflammatory substances such as AGEs and TNF-α may cross the BBB and directly enter the brain. In addition, low-molecular-weight substances such as amino acids and glucose can also cross the BBB, so AGEs may be produced in the brain if hyperglycemic conditions persist. In either case, it has been suggested that AGEs accumulated in the brain are associated with hippocampal atrophy [10]. Our previous studies have confirmed the association among AGE accumulation in the brain, hippocampal atrophy, and hyperglycemia-induced cognitive decline [10]. The receptor for AGE (RAGE) is known to play an important role in the development of inflammation in the brain [28]. AGEs generated by chronic hyperglycemia are recognized by RAGE, and AGEs themselves increase the expression of RAGE, promoting the production of intracellular oxidative stress and activating NF-κB [8,29,30]. Furthermore, hyperglycemia induces RAGE-dependent hippocampal spatial memory impairment [31], and systemic inflammation shifts the site of RAGE expression from endothelial cells to neurons in various parts of the brain [32]. Recently, it has been reported that hyperglycemia-induced inflammation in the hippocampus depends on the direct binding of RAGE expressed in microglia to serine/threonine protein kinase 1, and this mechanism is important in enhancing neuroinflammation in a hyperglycemic environment and is a key node in the development of cognitive impairment in diabetes [33]. In this study, it was found that the administration of RS reduced the Iba1 activation and TNF-α levels in the brain. RS is known to suppress inflammation in vascular endothelial cells [19] and systemic inflammation [20], and it is speculated that the ingestion of RS may suppress the expression of RAGE in organs, including the brain, and inhibit the accumulation of AGEs in the brain.

Another important anti-inflammatory effect of RS is to protect the glycocalyx layer of the vascular endothelium [20]. The glycocalyx layer is very fragile and easily destroyed during chronic inflammatory diseases such as arteriosclerosis, hyperglycemia, and diabetes [34,35,36,37]. In a hyperglycemic state, the synthesis of heparan sulfate and hyaluronic acid from glycosaminoglycans bound to the glycocalyx layer is inhibited and their detachment is promoted [32]. Mice administered MD are in a state of chronic inflammation [38]. Therefore, it is thought that the synthesis of heparan sulfate and hyaluronic acid is inhibited under inflammation, and that hyaluronidase secreted from inflamed endothelial cells degrades the glycocalyx layer of the pancreatic vascular endothelium [39,40]. We reported that RS administered to mice with LPS-induced inflammation protected the glycocalyx by inhibiting the degradation of syndecan-4, a component of glycosaminoglycans [20]. It has been reported that damage to the glycocalyx leads to the breakdown of the BBB, the inhibition of glycocalyx degradation maintains the integrity of BBB, and polysaccharides with anticoagulant properties such as heparin prevent the destruction of vascular endothelial glycocalyx [41]. Therefore, the administration of RS, which has anti-inflammatory properties [20] in addition to heparin-like anticoagulant properties [19], is thought to protect cerebrovascular endothelial cells from inflammation.

Regarding the mechanism of RS absorption in the body, because RS is a sulfated rhamnose polymer with a molecular weight of 50–300 kDa, it is difficult to transport directly to the brain, and instead, RS is absorbed into the bloodstream through intestinal M cells and exerts an anti-inflammatory effect on the vascular endothelium [20]. Regarding the mechanism by which RS suppresses inflammation in the brain, it is speculated that the anti-inflammatory effect of RS on the vascular endothelium reduces inflammatory cytokines secreted from the injured pancreas, which has been affected by diabetic inflammation, and that RS suppresses inflammation in blood vessels in the brain, thereby suppressing the activation of microglial cells in the brain tissue and suppressing the production of inflammatory cytokines. In addition, RS protects the glycocalyx layer of the vascular endothelium throughout the body, suppressing the migration of cytokines to the brain, preventing atrophy due to cell death in the hippocampal neuronal layer, and suppressing the decline in cognitive function.

In this study, RS did not show a clear improvement in behavioral and pharmacological tests, such as cognitive function, but showed a weak tendency for improvement in memory impairment. However, a clear improvement with RS was observed in physiological tests, such as the effect on blood glucose levels, and in pathological tests, such as hippocampal atrophy and changes in the expression of inflammation-related molecules. These results suggest that the continuous long-term intake of RS may improve cognitive function. Based on these results, a putative diagram of the process of brain inflammation and cognitive impairment caused by hyperglycemia and the mechanism by which RS administration suppresses brain inflammation and improves cognitive impairment are shown in Figure 10.

Epidemiological studies have shown that the incidence of Alzheimer’s disease is more than twice as high in people with type 2 diabetes than in those without diabetes [3,6]; however, it is unclear why patients with diabetes are more likely to develop Alzheimer’s disease. Insulin resistance observed in patients with diabetes leads to the accumulation of amyloid-β, the causative agent of Alzheimer’s disease [10]. In response to this, Wakabayashi et al. [42] revealed that, when type 2 diabetes develops due to the ingestion of a high-fat diet, the action of insulin in the blood is reduced, but the transfer of insulin to the brain is reduced, resulting in insulin resistance in the brain. They also showed that the amyloid-β clearance rate is reduced and its accumulation is increased in patients with diabetes. In fact, it has been suggested that various anti-diabetic drugs suppress the accumulation of amyloid-β [43], and a study by the American Academy of Neurology reported that people taking anti-diabetic drugs have a lower risk of Alzheimer’s disease than those not taking them [44]. In relation to this study, it is unclear whether hyperglycemia-induced brain inflammation leads to insulin resistance in the brain, slowing the removal rate of amyloid-β and causing amyloid-β to accumulate in the brain. Furthermore, it is unclear whether the administration of RS, which has the effect of improving hyperlipidemia [21], reduces the amount of amyloid-β accumulation in the brain. These questions require further investigation.

In this study, we were able to reveal that RS improves blood glucose levels in hyperglycemic mice, shows a tendency to improve memory function in behavioral pharmacological tests, inhibits atrophy of the neuronal layers in each region of the hippocampus, and suppresses inflammation in the brain. However, there are limitations to clarifying the causal relationship between these individual results, and they remain in the realm of speculation; thus, further investigation is required to determine the causal relationships between the results.

## 4. Materials and Methods

### 4.1. RS Sample

RS is composed of α-1,3-linked L-rhamnose residues, some of which are substituted with sulfate groups, mainly at the O-2 position, and contains traces of 1,2- and branched rhamnose residues [17]. Its molecular weight varies slightly depending on the production batch and ranges from tens to hundreds of thousands, but the average molecular weight of the RS used in this study was approximately 150 kDa [21]. In this study, Rhamnox^®^100, a health food ingredient manufactured and sold by Konan Chemical Manufacturing Co., Ltd. (Yokkaichi, Mie, Japan), was used as the RS sample. Rhamnox^®^100 is a hot water extract from the green alga *M. nitidum*. The raw green alga was treated with hot water and then filtered through a filter press to remove insoluble residues, and the extract was spray-dried into a powder. The product is specified to have an RS content of 70% or more. The RS content of the batch used in this study was determined to be approximately 75% based on the analytical procedure described in reference [21]. In this method, the RS content was calculated as the sum of three independently quantified components: constituent sugars, sulfate groups, and cations bound to sulfate groups. Constituent sugars were measured by high-performance liquid chromatography after acid hydrolysis and 1-phenyl-3-methyl-5-pyrazolone derivatization. Sulfate groups were quantified by ion chromatography after hydrochloric acid hydrolysis. Cations (Na^+^, K^+^, Mg^2+^, Ca^2+^) were measured by inductively coupled plasma optical emission spectrometry. The analyzed sample was found to contain 47% sugars, 22% sulfate, and 5.2% cations bound to sulfate groups.

### 4.2. Experimental Animals

A mouse model of type 2 diabetes was created by administrating nicotinamide (NA) (Wako Pure Chemicals, Tokyo, Japan) and streptozotocin (STZ) (Wako Pure Chemicals, Tokyo, Japan) to C57BL/6J mice (7 weeks old) (CLEA Japan, Tokyo, Japan) [45,46,47]. Mice were fasted for 24 h, after which 1.2% *w*/*v* NA was administered intraperitoneally at 240 mg/kg mouse body weight, followed 15 min later by 1% *w*/*v* STZ administered intraperitoneally at 100 mg/kg mouse body weight [45,46,47], followed by treatment with 1% maltodextrin (MD) (SLC, Shizuoka, Japan), which is a polysaccharide with 2–20 D-glucose sugar chains, for 4 months [10]. This procedure was repeated twice every other day to create a mouse model of type 2 diabetes [45,46,47]. Glucose loading was performed for 4 months to maintain a sufficiently high glycemic status compared to the control group. After this procedure, the MD-loaded group was fed a high-fat diet, DIO Rodent Purified Diet w/60% Energy From Fat/Blue (SLC, Shizuoka, Japan), and MD solution as drinking water. The DIO Rodent Purified Diet with w/60% Energy from Fat-Blue is a high-fat diet for rodents (60% of the energy comes from fat). The RS solution was prepared using Rhamnox^®^100 (Konan Chemical Manufacturing, Mie, Japan) containing 75% RS, diluted with tap water, or MD solution containing 75, 225, or 750 mg RS mouse body weight. Five groups of mice were created, with six mice in each group: a control group fed ad libitum with normal food and tap water, MD group, MD + RS (75 mg/kg) group (MD + RS75 group), MD + RS (225 mg/kg) group (MD + RS225 group), and MD + RS (750 mg/kg) group (MD + RS750 group). Body weights and blood glucose levels were measured monthly during the rearing period. Blood glucose levels were measured by drawing blood from the tail vein of the mice and measuring blood glucose levels using a human self-test glucose meter (Nipro Freestyle Freedom Lite) and a blood glucose-measuring electrode (Nipro FS blood glucose-measuring electrode Lite). The blood glucose level was set at an upper limit of 500 mg/dL because the upper limit of the measuring device was 500 mg/dL. The breeding method and schedule for the type 2 diabetes model used in this study are shown in Figure 11.

### 4.3. Behavioral Pharmacological Test

After a 4-month glucose tolerance test, spontaneous locomotor activity [48] was measured, followed by a rotarod test [49] and an eight-way maze test [50]. Spontaneous locomotor activity was measured using a locomotor activity measuring device (MERCQUEST Y. K, Toyama, Japan) that quantifies activity using an infrared sensor attached to the bottom of the cage [48]. The counter interface counted latticed infrared beam interruptions by a mouse moving in the cage inside the sensor. This count was measured for 20 min and was measured as spontaneous locomotion. In the rotarod test, after habituation on the experimenter’s palm(s) for more than 6 min, mice were placed on the stationary rod (d = 9 cm) of a rotating-rod device (Shin Factory, Fukuoka, Japan) for 120 s for adaptation to the experimental environment [49]. The mice were then carefully placed on the rod rotating at 10 round per minute (rpm), and the time they remained on the rod (retention time) was measured for each trial. The time taken for the mouse to fall into the lane was recorded as fall latency with a cut-off value of 360 s. One forced run was performed until the mouse fell or until 360 s elapsed. The eight-way maze (BioResearch, Nagoya, Japan) comprises an octagonal central platform (35 cm in diameter) with eight evenly spaced arms (60 cm long, 12 cm wide) extending outward, each capable of containing a food reward [50]. This configuration allows for the evaluation of the rodent’s ability to recall and distinguish between visited and unvisited arms, thereby measuring spatial memory and learning capacity. The eight-way maze test was performed using food as a reward for the fasting mice from the previous day. The MD solution administered to the glucose-loaded group was replaced with drinking water during fasting. The task completion time was defined as the time taken to finish eating the three foods, and the number of errors was counted as the number of times the mouse entered the room without eating the food, with a cut-off time of 10 min. The rotarod and eight-way maze tests were performed three times after 4 months of glucose loading; the first and second tests were used as training tests, and the third test was used as data [49,50].

### 4.4. Brain Sample Preparation

After performing the behavioral tests, the mice were anesthetized with barbital sodium salt solution (120 mg/kg). Brain tissue samples were fixed in 4% formaldehyde in phosphate-buffered saline (PBS) for 1 d, followed by dehydration and paraffin infiltration, after which the brains were embedded in paraffin blocks and sectioned to a thickness of 6 μm using Leica Microsystems (Wetzlar, Germany).

### 4.5. Histochemical and Immunohistochemical Analyses of Brain Samples

Histopathological changes in the brain were analyzed using hematoxylin–eosin (HE) staining. Immunohistochemical staining was performed to measure brain inflammation. The levels of inflammatory factor TNF-α was measured using anti-TNF-α antibody (Abcam ab1793, Cambridge, UK). Microglial activation was measured using an anti-Iba1 antibody (FUJIFILM 013-27691, Tokyo, Japan). The paraffin-embedded mouse brain sections were deparaffinized with xylene, hydrophilized with 100, 90, 80, 70, or 50% (*v*/*v*) ethanol, and immersed in PBS for 30 min. To further activate antigens, sections were immersed in 5% (*w*/*v*) urea and heated in a microwave oven. Endogenous peroxidase activity was blocked in 1% H_2_O_2_ in PBS, followed by immunohistochemical staining for 30 min in PBS containing 3% (*v*/*v*) normal goat serum antibody buffer. Rabbit polyclonal anti-TNF-α (2 mg/mL, 1:400) and anti-Iba1 (2 mg/mL, 1:400) antibodies were used as primary antibodies and incubated overnight at 16~18 °C [51]. Sections were washed with PBS and detected using a biotinylated horse anti-rabbit immunoglobulin kit (Vector Laboratories, Vectastain Elite ABC-HRP Kit, PK6101, and PK6102). Immune cells were detected using a 3,3′-diaminobenzidine peroxidase staining DAB kit (Nacalai Tesque, Kyoto, Japan) [51]. Morphological observations were performed using a microscope and a digital camera (BX53, DP74, Olympus, Tokyo, Japan). For the H&E-stained specimens, the thickness of the neuronal layer in the CA1, CA2, and CA3 regions and the dentate gyrus was measured at 10 points using ImageJ (National Institutes of Health, Bethesda, MD, USA), and the average of the 10 measured points was quantified. Stained samples were observed using microscopy. Iba1 and TNF-α levels were measured using the software ImageJ ver. 1.53 (National Institutes of Health, MD, USA) with a constant area, from six random fields of view. These regions are defined as ‘intensity’ in this study.

### 4.6. Statistical Analysis

Body weight, blood glucose, neuronal layer width in each hippocampal region, and behavioral pharmacological tests (motor activity, rotarod test, and eight-way maze test) were evaluated using one-way analysis of variance, followed by Dunnett’s method. Iba1 and TNF-α levels were also evaluated using the Dunnett method in the same manner. The statistical analysis software SPSS version 28 (IBM) was used for all data, and a significance level of less than 5% (*p* < 0.05) was considered significant.

## 5. Conclusions

The results of this study showed that the administration of RS may prevent cognitive decline in a hyperglycemic mouse model. The mechanism of action was suggested to be that PS suppresses the rise in blood glucose levels, inhibits the production of inflammatory factors such as TNF-α, inhibits the activation of glia cells, inhibits hippocampal atrophy, and improves hyperglycemia-induced cognitive impairment, including the repair of brain organs, in response to inflammatory cognitive impairment caused by hyperglycemia. These findings suggest that RS intake is useful in preventing brain decline and alleviating symptoms.

## Figures and Tables

**Figure 1 marinedrugs-23-00250-f001:**
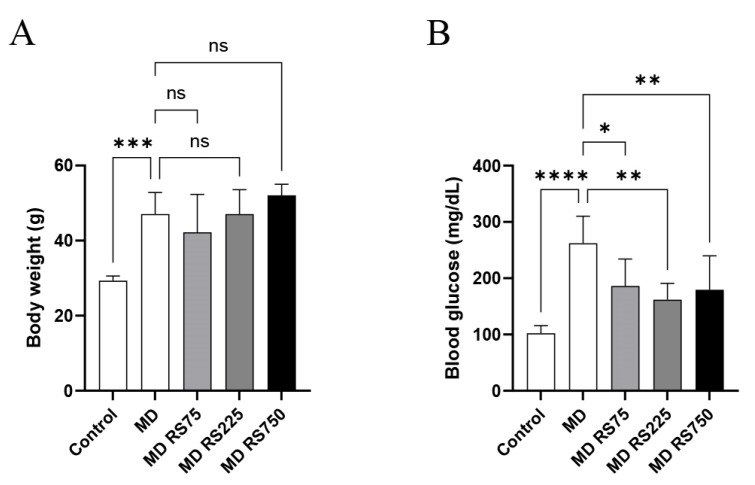
The effect of rhamnan sulfate (RS) administration on body weight and blood glucose level. (**A**) Body weight and (**B**) blood glucose level in each group 4 months after MD and/or RS administration are shown. Each value is expressed as mean ± SD (* *p* < 0.05, ** *p* < 0.01, *** *p* < 0.001, **** *p* < 0.0001). ns means not significant.

**Figure 2 marinedrugs-23-00250-f002:**
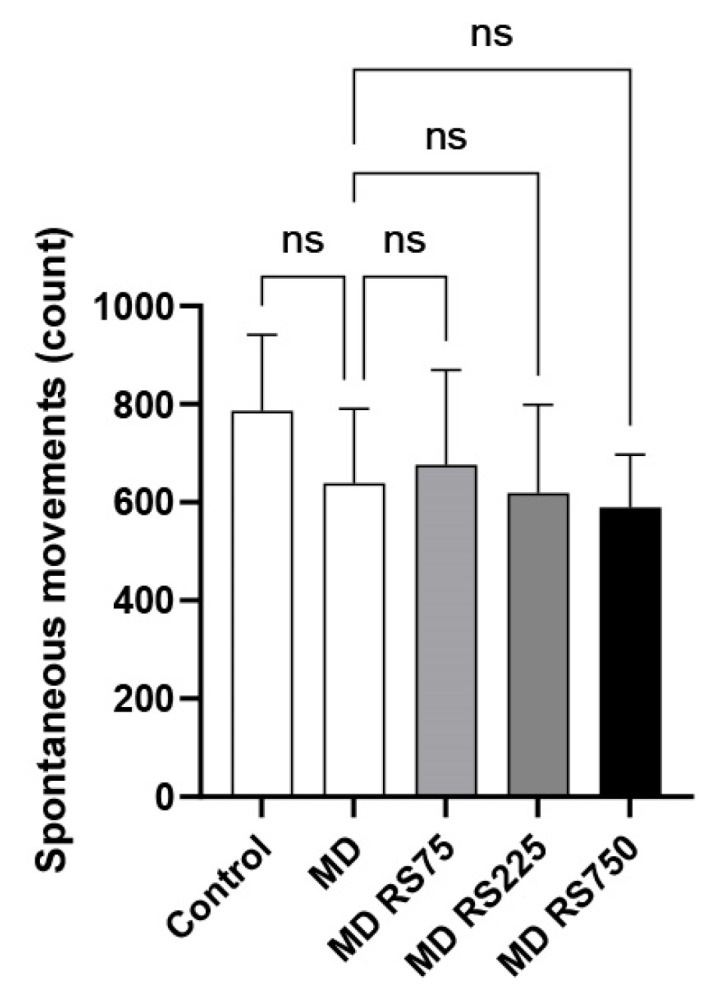
The effect of rhamnan sulfate (RS) administration on locomotion. After 4 months of MD loading, each mouse group was placed in a new environment, and 20 min of spontaneous locomotion was measured to compare the differences in locomotion. Each value is expressed as the mean ± SD. ns means not significant.

**Figure 3 marinedrugs-23-00250-f003:**
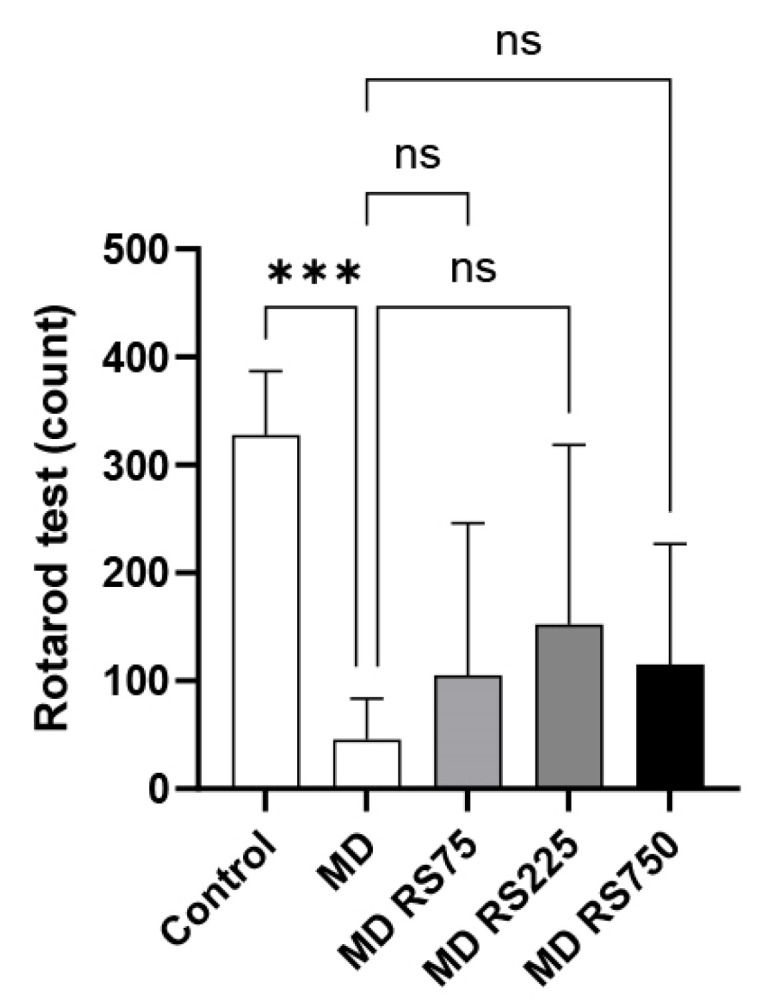
The effect of rhamnan sulfate (RS) administration on motor coordination (motor function). The upper limit was set at 6 min, and the time until the mice fell was measured to evaluate and compare motor and cognitive functions. Each value is expressed as the mean ± SD (*** *p* < 0.001). ns means not significant.

**Figure 4 marinedrugs-23-00250-f004:**
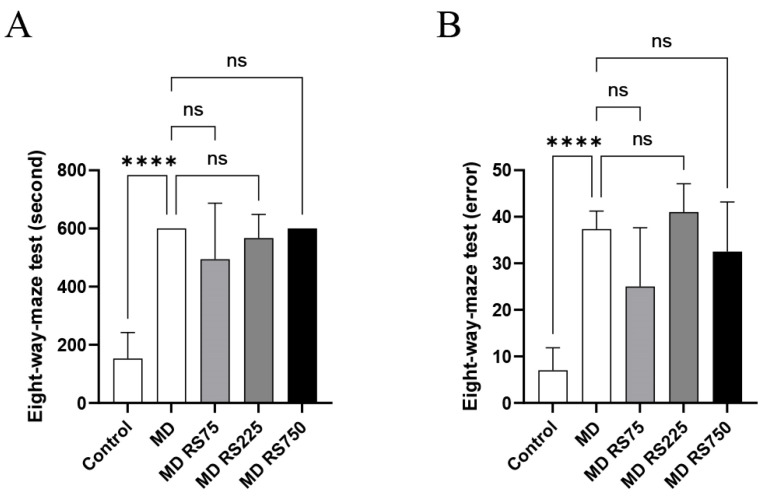
The effect of rhamnan sulfate (RS) administration on memory function. Cognitive memory function was assessed and compared by measuring (**A**) the time it took the mice to complete a task and (**B**) the number of errors they made while trying three rewards in 10 min. Each value is expressed as the mean ± SD (**** *p* < 0.0001 vs. control). ns means not significant.

**Figure 5 marinedrugs-23-00250-f005:**
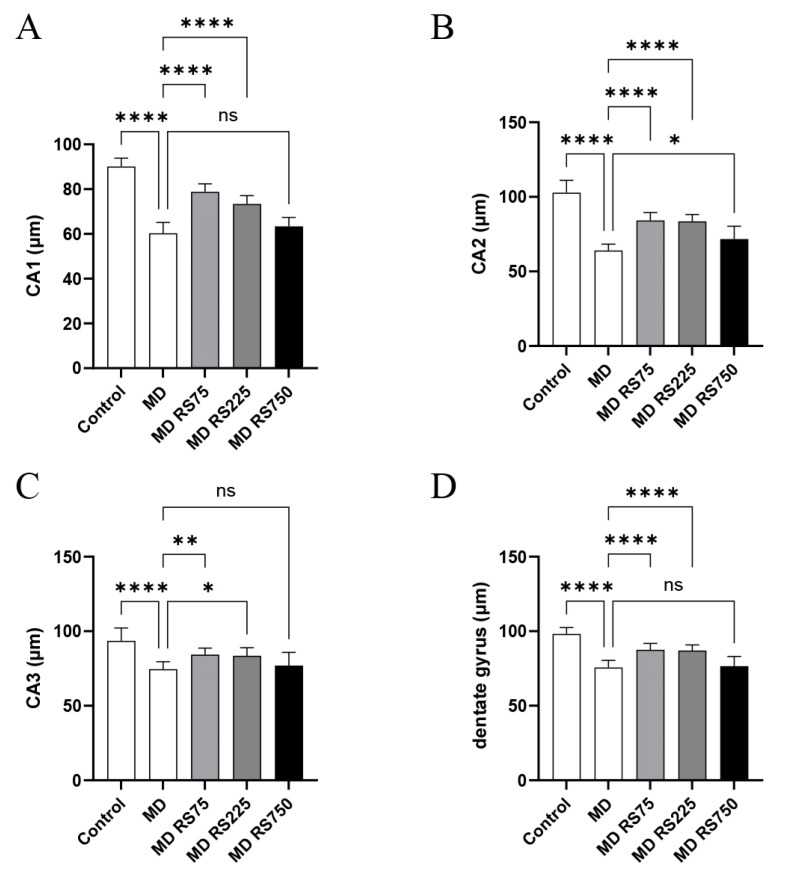
The effect of rhamnan sulfate (RS) administration on the hippocampal region (neuronal layer). RS was administered to a hyperglycemic mouse model, and its effects on the hippocampus (neuronal layer) were compared. The software ImageJ was used to calculate the average of 10 areas in the CA1, CA2, CA3, and dentate gyrus regions, and the results were quantified. (**A**): CA1 region, (**B**): CA2 region, (**C**): CA3 region, and (**D**): dentate gyrus region. Each value is expressed as the mean ± SD (* *p* < 0.05, ** *p* < 0.01, **** *p* < 0.0001). ns means not significant.

**Figure 6 marinedrugs-23-00250-f006:**
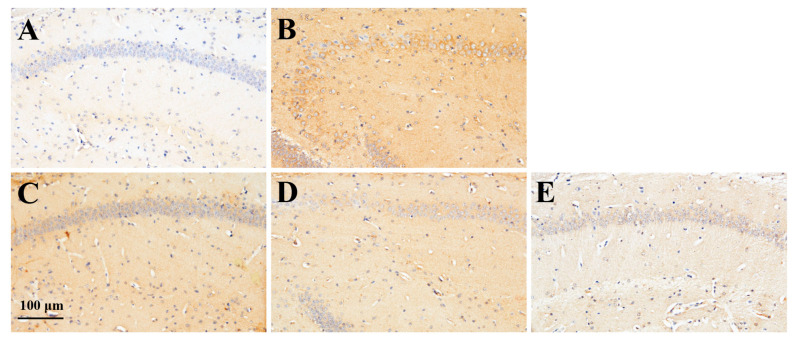
The effect of rhamnan sulfate (RS) on TNF-α expression in the brain of hyperglycemic mice. The stained images of TNF-α in each group ((**A**): control group, (**B**): MD group, (**C**): MD + RS75 group, (**D**): MD + RS225 group, and (**E**): MD + RS750 group). Scale bar: 100 μm.

**Figure 7 marinedrugs-23-00250-f007:**
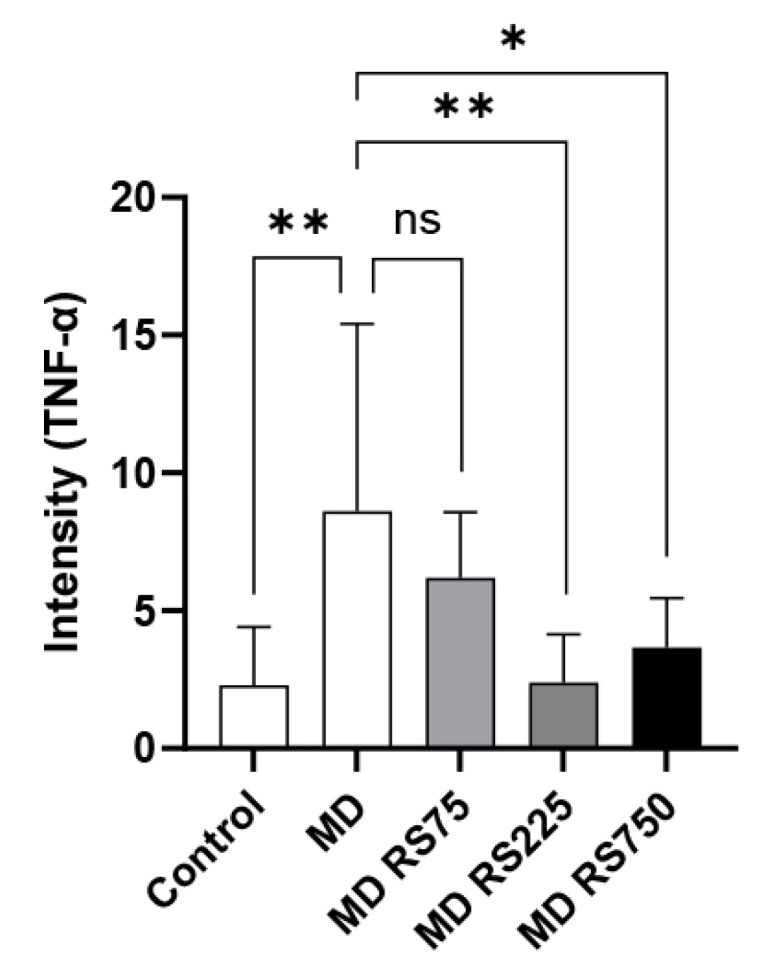
The effect of rhamnan sulfate (RS) intake on the TNF-α expression levels in the brain of hyperglycemic mice. The mean ± SD of five TNF-α staining images for each group shown in Figure 6 (control group, MD group, MD + RS75 group, MD + RS225 group, and MD + RS750 group) was calculated using the software ImageJ (* *p* < 0.05, ** *p* < 0.01). ns means not significant.

**Figure 8 marinedrugs-23-00250-f008:**
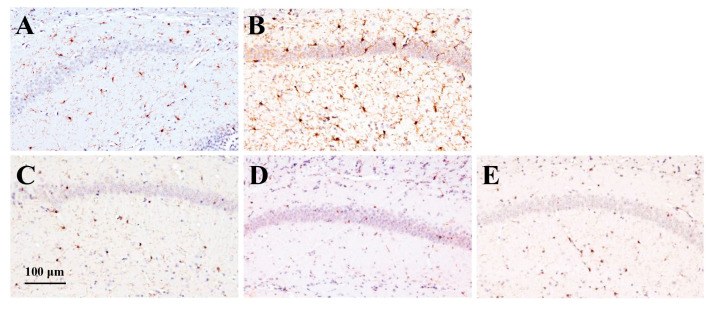
The effect of rhamnan sulfate (RS) on microglial activation in the brain tissue of hyperglycemic mice. The stained images of Iba1 in each group ((**A**) control group; (**B**) MD group; (**C**) MD + RS75 group; (**D**) MD + RS225 group; and (**E**) MD + RS750 group). Scale bar: 100 μm.

**Figure 9 marinedrugs-23-00250-f009:**
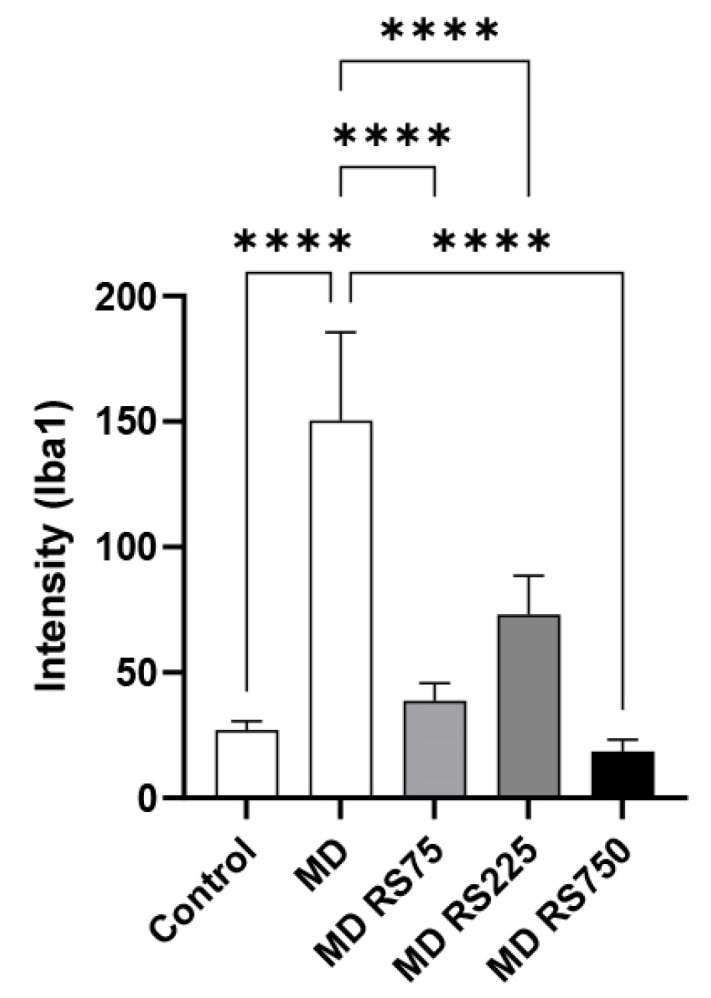
The effect of rhamnan sulfate (RS) intake on the Iba1 expression level. The mean ± SD of five Iba1 staining images for each group shown in Figure 8 (control group, MD group, MD + RS75 group, MD + RS225 group, and MD + RS750 group) was calculated using the software ImageJ (**** *p* < 0.0001).

**Figure 10 marinedrugs-23-00250-f010:**
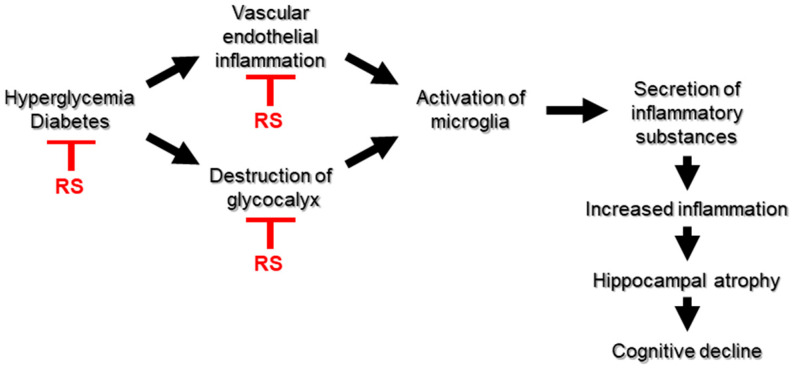
The hypothesized mechanism of the process of hyperglycemia-induced brain inflammation and cognitive decline, and the mechanism by which rhamnan sulfate (RS) suppresses brain inflammation and improves cognitive decline. This study suggests that RS suppresses the rise in blood glucose levels that leads to diabetes, inhibits vascular endothelial inflammation and glycocalyx damage, and thereby prevents microglial activation in the brain that leads to cognitive decline due to increased brain inflammation.

**Figure 11 marinedrugs-23-00250-f011:**
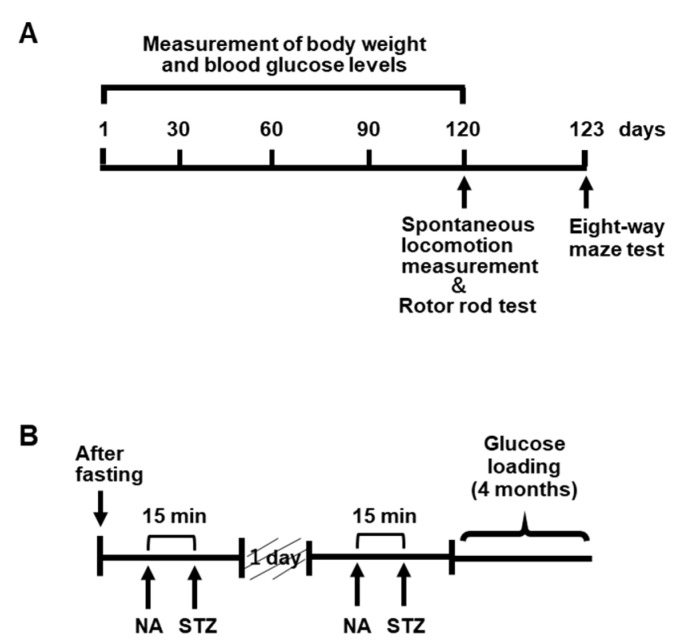
A protocol for creating a mouse model of type 2 diabetes and a four-month feeding schedule. (**A**) A hyperglycemic mouse model was established by pretreating mice with nicotinamide (NA) and streptozotocin (STZ) followed by glucose loading for 4 months. (**B**) Three behavioral pharmacological tests were performed after the 4-month glucose tolerance test: spontaneous locomotor activity, rotarod test, and eight-way maze test.

## Data Availability

The data presented in this study are available on request from the corresponding author.

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
