# Peer review of "Rhamnan Sulfate from the Seaweed *Monostroma nitidum* May Improve Cognitive Impairment Caused by Hyperglycemia"

_marinedrugs, 2025, doi:10.3390/md23060250_

Round 1
Reviewer 1 Report (Previous Reviewer 1)
Comments and Suggestions for Authors
The manuscript entitled “Rhamnan sulfate from the seaweed Monostroma nitidum may improve cognitive impairment caused by hyperglycemia” investigates whether rhamnan sulfate (RS) can alleviate hyperglycemia-induced cognitive decline in a mouse model. The study demonstrates that RS administration significantly reduced blood glucose levels in hyperglycemic mice and exhibited a trend toward improved cognitive function, as assessed by spontaneous locomotor activity, rotarod, and eight-arm maze tests. Although the behavioral improvements did not reach statistical significance, histopathological analysis revealed that RS treatment markedly attenuated neuronal layer atrophy across all regions of the hippocampus compared with untreated hyperglycemic controls. Moreover, RS significantly suppressed TNF-α expression and microglial activation in the brain. These findings suggest that RS may mitigate neuroinflammation and partially prevent cognitive impairment associated with hyperglycemic diabetes. However, several critical aspects warrant further clarification.
Comments
1. Dose-Efficacy Relationship: RS was administered at doses of 75, 225, and 750 mg/kg body weight. While all doses significantly lowered blood glucose levels, improvements in cognitive performance across the behavioral tests (spontaneous locomotor activity, rotarod, and eight-arm maze) were modest and did not achieve statistical significance. Notably, the effect was not dose-dependent, and the 225 mg/kg dose appeared to yield the most consistent improvements. This observation merits further investigation to determine the optimal therapeutic window and underlying pharmacodynamics.
2. Inflammatory Cytokine Migration to the Brain: In this hyperglycemic diabetic mouse model, it is hypothesized that chronic hyperglycemia leads to pancreatic β-cell death, with proinflammatory cytokines such as TNF-α and IL-1β being released and potentially migrating to the brain. However, the current study does not assess IL-1β levels or its distribution within the brain. Furthermore, the receptor for advanced glycation end-products (RAGE) is known to play a key role in neuroinflammation. The authors are encouraged to discuss the expression and role of IL-1β and RAGE in the brain, particularly in the context of hyperglycemia-induced cognitive dysfunction.
3. Microglial Activation and Downstream Pathways: Ionized calcium-binding adapter molecule 1 (Iba1) is a well-established marker for microglia and plays a critical role in the neuroinflammatory response and actin cytoskeleton remodeling in the central nervous system. The discussion should be expanded to further contextualize the significance of Iba1 expression in the present study. Additionally, the expression levels of related inflammatory signaling molecules—such as NF-κB, iNOS, and COX-2—should be examined or at least discussed to provide a more comprehensive understanding of the neuroinflammatory cascade and the potential mechanisms by which RS exerts its protective effects.
Author Response
Please see attachment.

Reviewer 2 Report (Previous Reviewer 2)
Comments and Suggestions for Authors
All issues were well addressed. I recommend it could be accepted for publication.
Author Response
Thank you for your valuable comments.
Round 2
Reviewer 1 Report (Previous Reviewer 1)
Comments and Suggestions for Authors
Accept in present form
Author Response
The manuscript entitled “Rhamnan sulfate from the seaweed Monostroma nitidum may improve cognitive impairment caused by hyperglycemia” investigates whether rhamnan sulfate (RS) can alleviate hyperglycemia-induced cognitive decline in a mouse model. The study demonstrates that RS administration significantly reduced blood glucose levels in hyperglycemic mice and exhibited a trend toward improved cognitive function, as assessed by spontaneous locomotor activity, rotarod, and eight-arm maze tests. Although the behavioral improvements did not reach statistical significance, histopathological analysis revealed that RS treatment markedly attenuated neuronal layer atrophy across all regions of the hippocampus compared with untreated hyperglycemic controls. Moreover, RS significantly suppressed TNF-α expression and microglial activation in the brain. These findings suggest that RS may mitigate neuroinflammation and partially prevent cognitive impairment associated with hyperglycemic diabetes. However, several critical aspects warrant further clarification.
Comment 1. Dose-Efficacy Relationship: RS was administered at doses of 75, 225, and 750 mg/kg body weight. While all doses significantly lowered blood glucose levels, improvements in cognitive performance across the behavioral tests (spontaneous locomotor activity, rotarod, and eight-arm maze) were modest and did not achieve statistical significance. Notably, the effect was not dose-dependent, and the 225 mg/kg dose appeared to yield the most consistent improvements. This observation merits further investigation to determine the optimal therapeutic window and underlying pharmacodynamics.
Response 1. Regarding the clinical dosage of RS in mice, the efficacy of RS has been verified in the past using various pathological model mice, and since the pathology was improved at about 300 mg/kg body weight (bw), the present experiment was performed at doses of 75, 225, and 750 mg/kg bw. As a result, at doses of 75 to 750 mg/kg bw, a significant inhibitory effect was observed on blood glucose levels and activation of microglia cells in the brain of hyperglycemic mice, at doses of 225 to 750 mg/kg bw, a significant inhibitory effect was observed on TNF-α expression in the brain, and at doses of 75 to 225 mg/kg bw, a significant inhibitory effect was observed on atrophy of brain hippocampal tissue in hyperglycemic mice.
Therefore, unlike chemical drugs, the absorption mechanism and mechanism of action of RS, a high molecular weight sulfated rhamnose polymer, in the body are complex, so the optimal dose is wide and may vary depending on the evaluation test method. Although a tendency for improvement in motor function and memory impairment was observed in the behavioral pharmacology test, no statistically significant difference was observed at any dose of RS, and it remains a possibility at this time. However, since improvement in brain tissue damage was observed depending on the duration of RS administration, it is expected that future research will clarify the cognitive impairment improvement effect of RS in more detail.
Comment 2. Inflammatory Cytokine Migration to the Brain: In this hyperglycemic diabetic mouse model, it is hypothesized that chronic hyperglycemia leads to pancreatic β-cell death, with proinflammatory cytokines such as TNF-α and IL-1β being released and potentially migrating to the brain. However, the current study does not assess IL-1β levels or its distribution within the brain. Furthermore, the receptor for advanced glycation end-products (RAGE) is known to play a key role in neuroinflammation. The authors are encouraged to discuss the expression and role of IL-1β and RAGE in the brain, particularly in the context of hyperglycemia-induced cognitive dysfunction.
Response 2. From the results of this study, it was speculated that hyperglycemic conditions in mice induce inflammation in mouse brain tissue and cause neurodegeneration (neuronal necrosis) in hippocampal nerves. Orally administered RS improves hyperglycemic conditions and suppresses inflammation in blood vessels, thereby suppressing the activation of microglial cells in the brain, suppressing the production off inflammatory cytokines such as TNF-α, and suppressing the degeneration of hippocampal nerve tissue caused by inflammatory cytokines. In this experiment, we did not measure IL-1β, because we want to show that RS administration suppresses the activation of microglial cells in the brain by measuring Iba1, a marker of microglial cell activation, and TNF-α, a representative inflammatory cytokine produced by microglial cells during inflammation.
On the other hand, inflammatory cytokines such as TNF-α and IL-1β, which are produced by inflammation of peripheral tissues such as pancreatitis, have large molecular weights and are therefore unlikely to pass through a normal blood brain barrier (BBB) and migrate directly to the brain. However, if the BBB is impaired, high molecular weight inflammatory substances such as AGEs and TNF-α may pass through the BBB and directly enter the brain. In addition, low molecular weight substances such as amino acids and glucose can also pass through the BBB, so AGEs may be produced in the brain when hyperglycemic conditions persist. In both cases, it has been suggested that AGEs accumulation in the brain is associated with hippocampal atrophy. In this case, it is thought that AGEs may cause neuroinflammation via the receptor for AGEs (RAGE). These points could not be proven in this study and require further verification. In relation to this, we cited a recent paper [CNS Neurosci Ther. 2024, 30: e14449.] and wrote the following in the discussion section of this paper about the possibility of RAGE-mediated neuroinflammation caused by AGEs generated in the brain under hyperglycemic conditions.
Recently, it has been reported that hyperglycemia-induced inflammation in the hippocampus depends on the direct binding of RAGE expressed in microglia to serine/threonine protein kinase 1, and this mechanism is important in enhancing neuroinflammation in a hyperglycemic environment and is a key node in the development of cognitive impairment in diabetes.
Comment 3. Microglial Activation and Downstream Pathways: Ionized calcium-binding adapter molecule 1 (Iba1) is a well-established marker for microglia and plays a critical role in the neuroinflammatory response and actin cytoskeleton remodeling in the central nervous system. The discussion should be expanded to further contextualize the significance of Iba1 expression in the present study. Additionally, the expression levels of related inflammatory signaling molecules—such as NF-κB, iNOS, and COX-2—should be examined or at least discussed to provide a more comprehensive understanding of the neuroinflammatory cascade and the potential mechanisms by which RS exerts its protective effects.
Response 3. Thank you for your valuable comments. We believe that the points you have made are very important issues that will enhance the significance of this study. We added the roles of Iba1 in the neuroinflammatory response and actin cytoskeleton remodeling in the central nervous system to the Discussion section. In future studies, we will also investigate the expression levels of related inflammatory signal molecules such as NF-κB, iNOS, and COX-2. We believe that the current data is insufficient to discuss how RS is involved in the neuroinflammatory cascade, and we would like to deepen the discussion in the process of confirming the effect of RS on the expression levels of NF-κB, iNOS, and COX-2, etc.
In this study, we were able to clarify that brain tissue and brain function are impaired in a hyperglycemic state, and we were able to demonstrate the possibility that administration of RS, a seaweed component that is ingested daily, can improve brain tissue damage and improve brain dysfunction caused by a hyperglycemic state. Therefore, we believe that publishing this study as soon as possible is academically significant. In the future, we would like to take the time to investigate the more detailed molecular mechanism of the cognitive dysfunction improvement effect of RS.
This manuscript is a resubmission of an earlier submission. The following is a list of the peer review reports and author responses from that submission.
Round 1
Reviewer 1 Report
Comments and Suggestions for Authors
Comments:
The manuscript describes "Rhamnan sulfate from the seaweed Monostroma nitidum may improve cognitive impairment caused by hyperglycemia” To investigate whether rhamnan sulfate (RS) extracted from the seaweed Monostroma nitidum could improve diabetic brain dysfunction in a hyperglycemic mouse model induced by nicotinamide and streptozotocin pretreatment followed by oral administration of a high-fat diet and maltodextrin (MD) for 4 months. Analysis showed that compared with the untreated hyperglycemic control group, RS significantly reduced the atrophy of neuronal layers in various regions of the hippocampus, inhibited the expression of TNF-α and activation of microglia in the brain, and showed a trend to reduce the accumulation of advanced glycation end products in the brain. The results suggest that RS intake may suppress brain inflammation and alleviate cognitive impairment associated with hyperglycemic diabetes., but several points need clarification.
Comment:
- The animal behavior experiments in this article are all invalid. Should the dosage be lowered? Lower doses (e.g., 10–50 mg/kg) are then added to determine the minimum effective dose.
- Whether to add tests such as Morris water maze (spatial memory) and novel object recognition (short-term memory) to comprehensively evaluate the impact of RS on different cognitive domains.
- Whether to supplement the forced swimming test or elevated plus maze was to explore the improving effect of RS on anxiety/depression associated with diabetes.
- The animal experiments in this article only involved anti-inflammatory tissue staining in the brain. What is the mechanism? Whether RS penetrates the BBB can be detected by mass spectrometry or fluorescent labeling, and its brain distribution concentration can be quantified to confirm direct action or indirect (such as through the gut-brain axis) mechanism.
Comments on the Quality of English Language
The English could be improved to more clearly express the research.
Reviewer 2 Report
Comments and Suggestions for Authors
This study systematically discussed the intervention mechanism of fucose sulfate (RS) on hyperglycemic-induced cognitive impairment. The design of the paper is reasonable, and the data preliminarily support the improvement effect of RS on hyperglycemic-related cognitive impairment. However, it is necessary to revise the expression of results, enhance statistical rigor, and supplement the mechanism discussion.
- The summary and conclusion sections of the manuscript describe behavioral pharmacological tests results as "tendency to improve" but do not indicate significance (p-value range).
- Lines 27-29, only the RS100 group showed a marginal trend in the accumulation of AGEs (p=0.076), and it should be noted that there were no significant differences in the other dosage groups.
- Table 1-3, should the total number of samples be 30 mice?
- Table 1-3, how does the author define +1, +2, +3? Please add in the Methods section.
- The results showed that RS inhibited TNF-α expression and microglial cell activation, but the mechanism was not further discussed and verified.
- Why did the authors choose the injection doses of 100, 300, 1000mg/kg?
- Line 138, the description here does not match what is shown in Figure 5; p<0.0001 is not shown in Figure 5C.
- Can the model used in the manuscript fully simulate the chronic high sugar state in humans?
Reviewer 3 Report
Comments and Suggestions for Authors
This manuscript presents an interesting investigation into the potential neuroprotective effects of rhamnan sulfate (RS) from Monostroma nitidum in a hyperglycemic mouse model. The study is well-structured with a clear research question and methodology. The authors have conducted comprehensive experiments to evaluate both behavioral and histopathological changes, which strengthens their findings.
The connection between diabetes and cognitive impairment is clinically relevant, and exploring natural compounds as potential therapeutic agents is a valuable approach. The authors' work on RS builds upon their previous studies and provides new insights into its effects on brain inflammation and cognitive function.
Major Issues Requiring Revision
- Statistical Analysis Inconsistencies (Lines 438-444): The statistical methods are inadequately described. The authors state they used Dunnett's test and one-way ANOVA, but do not clearly specify which test was used for which data set. Additionally, for multiple comparisons after ANOVA, the post-hoc test should be explicitly mentioned. The chi-square test for immunohistochemical scoring requires more justification as this may not be appropriate for small sample sizes (n=6 per group).
- Contradictory Results Interpretation (Lines 235-239): Authors state "no significant differences were observed in the 8-way maze test" but later claim "RS administration may improve memory and cognitive function in tasks." This contradictory interpretation weakens the conclusions. Without statistical significance, such claims should be tempered or more clearly presented as preliminary observations.
- Methodological Limitations in Animal Model (Lines 349-361): The manuscript fails to adequately justify why this specific model (nicotinamide + streptozotocin followed by high-fat diet + maltodextrin) was selected over established models of diabetic cognitive impairment. The authors cite their previous work (references 10-12) but these are published in a non-indexed journal that is "currently unavailable to anyone other than members of the Japanese Society of Psychiatric Pharmacology" (lines 477-481), making validation impossible for most readers.
- Inadequate Discussion of Mechanism (Lines 525-300): While the authors propose that RS affects brain inflammation via effects on the vascular endothelium, the mechanism of how RS (a large molecular weight polysaccharide, 50-100 kDa) could affect the brain is insufficiently explained. Line 297 states "it is difficult to transport directly into the brain" but the subsequent explanation about absorption "via intestinal M cells" does not adequately address how RS or its metabolites might cross the blood-brain barrier.
- Overinterpretation of Histopathological Findings (Lines 169-180 and 187-197): The scoring system used for TNF-α and Iba1 immunostaining (0 to +3) appears subjective, and the authors do not mention blinding in their assessment, raising concerns about potential bias. Additionally, the images in Figure 6 and 7 show selected areas, but the quantification method of "10 measured points" (line 437) is not clearly explained in terms of how these points were selected across different brain regions.
These significant issues undermine the reliability and novelty of the findings. While the topic is promising, substantial revisions addressing these methodological and interpretational concerns would be necessary before the manuscript could be considered acceptable for publication.